# Health Impacts of Ambient Biomass Smoke in Tasmania, Australia

**DOI:** 10.3390/ijerph17093264

**Published:** 2020-05-07

**Authors:** Nicolas Borchers-Arriagada, Andrew J. Palmer, David M.J.S. Bowman, Grant J. Williamson, Fay H. Johnston

**Affiliations:** 1Menzies Institute for Medical Research, University of Tasmania, Hobart 7000, Tasmania, Australia; nicolas.borchers@utas.edu.au (N.B.-A.); andrew.palmer@utas.edu.au (A.J.P.); 2New South Wales Bushfire Risk Management Research Hub, University of Tasmania, Hobart 2522, Tasmania, Australia; 3Centre for Health Policy, School of Population and Global Health, The University of Melbourne, Parkville 3010, Victoria, Australia; 4School of Natural Sciences, University of Tasmania, Launceston 7250, Tasmania, Australia; David.Bowman@utas.edu.au (D.M.J.S.B.); Grant.Williamson@utas.edu.au (G.J.W.)

**Keywords:** woodsmoke, fires, fine particulate matter, health effects, asthma, health costs, woodstoves

## Abstract

The island state of Tasmania has marked seasonal variations of fine particulate matter (PM_2.5_) concentrations related to wood heating during winter, planned forest fires during autumn and spring, and bushfires during summer. Biomass smoke causes considerable health harms and associated costs. We estimated the historical health burden from PM_2.5_ attributable to wood heater smoke (WHS) and landscape fire smoke (LFS) in Tasmania between 2010 and 2019. We calculated the daily population level exposure to WHS- and LFS-related PM_2.5_ and estimated the number of cases and health costs due to premature mortality, cardiorespiratory hospital admissions, and asthma emergency department (ED) visits. We estimated 69 deaths, 86 hospital admissions, and 15 asthma ED visits, each year, with over 74% of impacts attributed to WHS. Average yearly costs associated with WHS were of AUD$ 293 million and AUD$ 16 million for LFS. The latter increased up to more than AUD$ 34 million during extreme bushfire seasons. This is the first study to quantify the health impacts attributable to biomass smoke for Tasmania. We estimated substantial impacts, which could be reduced through replacing heating technologies, improving fire management, and possibly implementing integrated strategies. This would most likely produce important and cost-effective health benefits.

## 1. Introduction

Smoke from biomass combustion, including wood heater smoke (WHS) and landscape fire smoke (LFS), is composed of a complex blend of pollutants such as particulate matter, carbon monoxide, and volatile organic gases [1,2]. WHS is produced by emissions from a myriad of residential heating technologies such as wood or pellet stoves, biomass boilers, and open fireplaces. There is a great variation in the physicochemical properties of particles that are emitted, and they depend on the type of technology, fuel conditions, and fuel types, among others [3]. Likewise, the composition of LFS varies according to vegetation type, climate conditions, and intensity of burn [4]. Short- and long-term exposure to particulate matter, specifically the fine fraction that contains particles of size up to 2.5 μm (PM_2.5_), has been clearly linked to several health problems, including premature mortality, cardiovascular (CVD) and respiratory (RSP) hospital admissions, and emergency department (ED) visits [5]. Multiple studies have investigated these exposure–outcome relationships and estimated the health impacts, particularly for WHS and LFS [6,7,8,9,10,11,12]. In general, population exposure to outdoor WHS is seasonal with the highest concentrations observed during the winter months, especially during nocturnal temperature inversions that inhibit dispersal of the smoke. In contrast, LFS is a more sporadic source typically occurring in warmer months. Population exposure to LFS is usually for much shorter duration than WHS, i.e., for days rather than months at a time, but peak concentrations of PM_2.5_ can be considerably higher. Accordingly, the public health impacts are different. For instance, Bowman et al. [13] estimated that an extreme fire event in Chile that lasted more than 16 days resulted in an average PM_2.5_ increase of 26.8 μg/m^3^ leading to 76 premature deaths and 209 respiratory and cardiovascular hospital admissions from an exposed population of more than 9.5 million. In the US, Fann et al. [14] estimated that for the period 2008–2012 health costs of short-term exposure to wildfire smoke PM_2.5_ ranged between US$ 11 and US$ 20 billion per year, while costs associated with long-term exposure ranged between US$ 76 and US$ 130 billion per year (2010 $US). These authors observed that a relatively small number of states were highly affected by fires during this period, and that the population-weighted annual mean LFS-attributable PM_2.5_ ranged from 0.6 to 1.1 μg/m^3^ [14]. In contrast, Sarigiannis et al. [15] estimated that for a population of 900,000 in Greece, 200 premature deaths per year could be attributed to WHS in the cold season, with an estimated cost of between 200 and 1200 million Euros. For Australia, Robinson [16] estimated that WHS results in annual health costs of AUD$ 3800 per woodstove.

The temperate island state of Tasmania is located south of continental Australia, between 40°S and 44°S, and is characterized by having the coldest temperatures in Australia. With a population of 522,000 (in 2017), its urban population is concentrated in the north–east, east, and south–east, mainly in four cities: Hobart, Launceston, Devonport, and Burnie [17]. In contrast, the rugged and sparsely settled interior and west of the island is mostly wilderness with few settlements. The high western mountains and central plateau regularly experience freezing temperatures throughout the winter months with occasional snowstorms. The vast majority of the islands’ air pollution is due to biomass smoke from domestic wood heaters in winter and, to a lesser extent, from planned burns during autumn [18,19,20,21]. Poor air quality is an important concern, mainly during colder months, due to WHS [22,23]. With an estimated number of 69,000 wood heaters [19], these remain the main source of energy used for heating purposes for around 30% of the population. During warmer months, air pollution in populated areas is consistently very low, except when wildfires or deliberate management burns are conducted in forested areas, which can sometimes cause large amounts of smoke to affect densely populated areas. Planned landscape fires for fuel and forestry management are concentrated in spring and autumn months and wildfires in summer.

Previous studies have used Tasmania as a setting and have characterized air pollution [20,21,24,25,26], assessed public health interventions to reduce WHS emissions [23], and quantified the association between biomass smoke and health outcomes [7,22]. No study has quantified the health burden attributable to biomass smoke-related PM_2.5_ in Tasmania, and compared the relative contribution of LFS and WHS. We used publicly available demographic health and air quality data to estimate and compare health impacts and health costs attributable to the two most important sources of air pollution in Tasmania, wood heaters and landscape fires, between 2010 and 2019.

## 2. Materials and Methods

### 2.1. PM_2.5_ Exposure and Identification of WHS and LFS Days

The state of Tasmania has the advantage of a long-standing dense network of air quality monitoring stations maintained by the Environment Protection Authority (EPA Tasmania), with more than 89% of the population living within 5 km of a monitoring station. Hourly PM_2.5_ records were obtained from Tasmania EPA for all Base Line Air Network of EPA Tasmania (BLANkET) monitoring stations between January of 2010 and December of 2019 [27]. Daily averages were estimated when at least 18 valid hourly records (larger than zero) were available. Historical minimum and maximum temperature data was obtained from the Bureau of Meteorology (BOM) [28], for all meteorological stations active between January 2010 and December 2019. Average heating degree days (HDD), sum of degrees Celsius for which the average daily temperature was below a theoretical comfort temperature of 18 °C, was estimated for each station. The larger the daily HDD, the higher the probability of having daily low temperatures and increased use of domestic heating. Daily PM_2.5_ exposure, daily HDD, minimum daily temperature, and maximum daily temperature were interpolated at a Statistical Area Level 2 (SA2), a geographical area defined by the Australian Bureau of Statistics [29], that is characterized by having an average population of 10,000, ranging between 3000 and 25,000 persons. We used an inverse distance weighting (IDW) method [30], a spatial interpolation algorithm that uses observations at known locations (e.g., air quality at monitoring stations) to calculate unknown values at other places by giving more importance (weight) to known values that are closer compared to those that are farther away. We estimated daily averages (PM_2.5_, HDD, temperature) at each SA2 by considering only air quality monitoring stations or BOM meteorological stations that were within a 100 km radius from the SA2 centroid. Tasmania’s relatively dense air quality monitoring network consisted of 17 monitors active in 2010 and 35 in 2019. Likewise, the BOM meteorological network in Tasmania was composed of 58 active monitors in 2010 and 57 in 2019. A map with the location of all BLANkET and BOM stations may be seen in Appendix A.

The background (or counterfactual) PM_2.5_ concentration was estimated as the average PM_2.5_ for summer months, excluding days when it was likely that a landscape fire happened. For each SA2, we identified days with likely landscape fire activity as those summer days (November to February) when daily PM_2.5_ was above the 95th percentile of historical PM_2.5_ daily averages for each SA2. This threshold has been used to identify most probable fire smoke days in previous studies [31,32,33]. Whenever the daily PM_2.5_ was higher than the estimated counterfactual, we estimated the attributable PM_2.5_ portion using the following equation:
PM_2.5_attributable,sa2__ = PM_2.5_daily,sa2__ − PM_2.5_counterfactual,sa2__

Days in which the daily PM_2.5_ was less than or equal to the estimated counterfactual, were defined as unpolluted. Other days were defined as either primarily WHS- or LFS-affected using the approach described below.

Air pollution from wood heaters and landscape fires has characteristic seasonal and daily temporal patterns which make discerning the source of air pollution in Tasmania straightforward for the summer and winter periods (see Appendix A). Ambient PM_2.5_ in Tasmania is highly dominated by biomass (wood heater and landscape fires) smoke, and in some locations less than 8% would be attributable to other sources such as vehicle emissions, local industry, and other sources of fine aerosols [20]. Air pollution generated by wood heaters follows a common pattern throughout the cooler winter months (May to August), with characteristic seasonal and diurnal patterns with a large peak overnight, and smaller peak in the early morning [18,19,20,23]. However, the transition months during autumn and spring potentially have both sources, depending on daily weather conditions that might either favor wood heater use, or landscape fires. For transition months (March, April, September, October), we predicted the most probable source by using a machine learning algorithm known as random forest. This type of algorithm applies random sampling over a set of observations with known categories or classifications to train a model, and later uses this model to predict over observations with unknown categories [34]. We trained a model using known source categories during summer (LFS) and winter (WHS), and applied it to days during transition months using the following explanatory variables: geographic location by statistical area (SA2), year, month, day, daily PM_2.5_ average, daily HDD average, day of the week, minimum daily temperature, and maximum daily temperature.

We evaluated the sensitivity of our results to the following assumptions:1)The PM_2.5_ threshold used to identify LFS summer days (90th vs 99th percentile);2)The months considered as start and end of winter; and3)The consideration of sources allocated during the transition months (March, April, September, October) through the random forest method.

### 2.2. Population and Health Data

Estimated resident population data by sex and age group was obtained from the Australian Bureau of Statistics [17]. State-wide all-cause death counts by age and sex were obtained from the Australian Bureau of Statistics [35]. Respiratory and circulatory disease hospitalization incidence rates for the Tasmanian population were estimated using the online tables from the Aboriginal and Torres Strait Islander Health Performance Framework 2017 [36]. Asthma ED visit counts were obtained from the Australian Institute of Health and Welfare [37,38,39]. Yearly averages for population and health data by age group and sex, where available, are presented in Appendix A. Tasmania-wide annual average base incidence rates for mortality and ED visits were estimated by dividing the number of cases by population.

### 2.3. Health Impacts

We estimated the number of premature deaths, respiratory and cardiovascular hospital admissions, and asthma ED visits between January 2010 and February 2019 using standard methods for a health impact assessments (HIA) [40].

Cases were estimated using the following equation:
Cases = IR × Pop × (e^β×ΔC^ − 1) ≅ IR_o_ × Pop × β_o_ × ΔC
where Cases is the total number of estimated cases, IR is the base incidence rate, Pop is the total estimated exposed population, β is the health outcome risk coefficient, and ΔC is the change in PM_2.5_ concentration. Annual average IR and ΔC were used for estimating long-term effects and 24-h (daily) averages for short-term effects.

Timeframes of exposure to WHS-related PM_2.5_ and LFS-related PM_2.5_ are different by nature, with population exposure to WHS happening every year throughout winter months, and exposure to LFS happening sporadically and during a shorter duration (i.e., days) mainly during summer months. Given this, we selected different dose–response functions to assess the impact of smoke exposure on premature mortality. In the case of WHS, we assessed long-term impacts using average annual exposure, for which the relationships have been characterized [5]. For LFS, we estimated impacts on premature mortality by using average daily exposure, as there is no available evidence on the association between premature mortality and long-term exposure to LFS [41], and this a sporadic rather than a chronic phenomenon in Tasmania. For hospital admission and ED visits, we used average daily exposure [5,11,42]. We selected the health coefficients presented in Table 1, and considered uncertainty associated with selected coefficients, to obtain the health impacts’ 95% confidence intervals.

### 2.4. Health Costs and Assessment

All health impacts were valued using accepted environmental and health economics methods [43,44]. For mortality, we considered the Value of Statistical Life (VSL) as AUD$ 4.2 million (2014 AUD$), as per recommendations from the Office of Best Practice and Regulation [45]. We estimated hospitalization costs using a cost of illness (COI) method, considering two elements: (1) direct health care costs; and (2) indirect lost productivity due to hospitalization days, measured as lost salary. Average health care (hospital) costs and length of stay were estimated using the Independent Hospital Pricing Authority [46] National Cost Data Collection Cost Report. Average daily salary was estimated using the Average Weekly Earnings and Labour Workforce Statistics for Tasmania published by the Australian Bureau of Statistics [47,48]. We obtained an average hospitalization cost of AUD$ 7193 (2016 AUD$) and AUD$ 7280 (2016 AUD$) per case for circulatory and respiratory diseases, respectively. ED visits were valued considering average health care costs using the Health Policy Analysis [49] Emergency Care Costing Report, with an estimated AUD$ 705 (2016 AUD$) per case. All costs were adjusted by inflation to Australian Dollars of 2018, using values recommended by the Reserve Bank of Australia [50].

Translating these costs to indicators (see Appendix A) helps inform policy. Accordingly, we estimated average daily costs for WHS and LFS, only considering the respective number of days in which either WHS or LFS were identified. To obtain average yearly WHS cost per woodstove, we estimated a total of 69,317 woodstoves for Tasmania, using raw survey data obtained from EPA Tasmania [19] and the number of dwellings per mesh block obtained from Australian Bureau of Statistics [51] (see Appendix A).

## 3. Results

### 3.1. PM_2.5_ and HDD

Consistent with previous research in Tasmania we observed clear seasonal and geographic patterns in PM_2.5_ concentrations and HDD that reflect the island geography (Appendix A). Figure 1 shows average PM_2.5_ concentrations and HDD for summer, transition and winter months by SA2, together with population density. During winter, there were increases in HDD and PM_2.5_ concentrations, with lower values seen during summer months. A slight decrease in PM_2.5_ was observed during transition months, probably due to the lower presence of wood heater smoke.

Time trends for population-weighted PM_2.5_ concentration and HDD demonstrated a clear association that was cyclical increasing during winter months and decreasing during summer months. There were exceptions, however, particularly the summers of 2013, 2014, 2016, and 2019, when the presence of major fires lead to state-wide daily PM_2.5_ averages reaching 34.2 μg/m^3^, 16.5 μg/m^3^, 59.7 μg/m^3^, and 48.6 μg/m^3^ (Figure 2). Summary statistics by BLANKeT station and attributed PM_2.5_ fractions for LFS and WHS per month are presented in the Appendix A.

Unpolluted days occurred throughout the year and had average values below 2 μg/m^3^ and maximum values of 3 μg/m^3^ of PM_2.5_; WHS days had slightly higher 24-h PM_2.5_ averages, but LFS had a greater variation and higher maximum values (Table 2; detail by year presented in Appendix A).

### 3.2. Health Impacts

During the period of analysis, we estimated (Table 3) that biomass smoke was responsible for 688 premature deaths (95% confidence interval (CI): 433–932), 857 hospital admissions (95% CI: 62–1725), and 148 asthma ED visits (95% CI: 74–229). Over 74% of the morbidity impacts and 94% of the mortality impacts were attributed to WHS. This difference is closely related to the nature of exposure: long-term in the case of WHS and short-term in the case of fires.

As expected, cases attributable to WHS were concentrated in winter with the total number of cases peaking in June (Figure 3). On average, the number of cases attributable to LFS was mostly concentrated in January, followed by February, April and October. Unlike WHS, LFS health impacts were not similarly distributed from year to year, but varied according to the intensity of the fire seasons, with particularly high number of cases during January of 2016 and 2019.

For 2016 and 2019 we estimated eight (95% CI: 3–13) premature deaths, 18 (95% CI: 9 –31) asthma ED visits, 15 (95% CI: 3–28) cardiovascular hospital admissions, and 24 (95% CI: 0–52) respiratory hospital admissions (Table 4). The exclusion of those two years makes average yearly number of cases drop to three (95% CI: 1–4), five (95% CI: 3–8), five (95% CI: 1–9), and eight (95% CI: 0–17) for all-cause mortality, asthma ED visits, cardiovascular hospital admissions, and respiratory hospital admissions, respectively.

### 3.3. Health Costs

We estimated a total AUD$ 161 (95% CI: 58–264) million attributable to LFS, and AUD$ 2934 (95% CI: 1885–3930) million attributable to WHS (Table 5). This translates into average yearly costs of AUD$ 16 (95% CI: 6–26) million and AUD$ 293 (95% CI: 189–393) million for LFS and WHS, respectively. With these estimates, 5.2% of yearly smoke-related health costs are attributable to LFS and 94.8% to WHS. For 2016 and 2019 together, LFS was responsible for 12.1% of smoke-related health costs, while WHS for 87.9%.

The distribution of health costs varied considerably by region (SA4 level) with Hobart being the most impacted, followed by Launceston and north–east (Table 6). While distribution of costs for LFS reflected population distribution, we observed a higher distribution for the area of Launceston and north–east in the case of WHS.

We had previously identified that the years 2016 and 2019 had particularly important health impacts attributed to LFS, and we estimated that the average yearly health costs during those years increased up to AUD$ 34.5 (95% CI: 12.4–57.2) million for 2016 and AUD$ 34.5 (95% CI: 12.4–57.1) million for 2019 (Figure 4). Other years like 2011 and 2018 had relatively lower health costs of AUD$ 7.4 (95% CI: 2.7–12.1) million and AUD$ 9.2 (95% CI: 3.4–15) million, respectively.

Table 7 shows that on average the unitary impacts of WHS may be summarized as AUD$ 1.57 million /WHS-day, while for LFS we estimated an average AUD$ 75,954 /LFS-day. Furthermore, the average health burden attributable to one woodstove is of AUD$ 4232 /woodstove-year.

Although average daily costs for WHS were considerably higher than those for LFS, particularly severe LFS days produced substantially higher daily health costs of more than AUD$ 4 million, which was well above the average daily cost of WHS (see Appendix A).

### 3.4. Sensitivity Analysis

Table 8 provides results for the different health economic indicators (defined in Appendix A). We present two broad groups, one including all months, and the other excluding months which had their pollution source predicted through a random forest algorithm. We present the range of variation for the selected indicators as a result of varying the PM_2.5_ threshold used to identify LFS summer days, and the months used to define summer and winter.

Costs attributable to LFS vary considerably between AUD$ 13.8 million and AUD$ 27 million per year, equivalent to between AUD$ 64,000 and AUD$ 109,000 per LFS-day. The lower variation in the average per day costs is due to the inclusion of a lower number of LFS days in the lower cost scenario. The lowest costs were estimated when the 99th percentile of historical PM_2.5_ daily averages was used as a threshold to identify LFS summer days and winter was defined between May and July. On the other hand, the highest costs were estimated when the 75th percentile was used to define LFS summer days and winter only included June and July. In the case of WHS, results were less sensitive, ranging between AUD$ 245.8 million and AUD$ 318.9 million, equivalent to between AUD$ 1.4 million to AUD$ 1.7 million per WHS-day, or between AUD$ 3545 and $4600 per woodstove-year. The highest cost was obtained when threshold for identifying an LFS summer day was the 75th percentile, and winter included months between May and July. The lowest WHS costs were estimated when we used the 99th percentile threshold for LFS identification, but winter was only defined by June and July. When excluding months with predicted biomass smoke source total and yearly costs were reduced by 17% and 39% for WHS and LFS, respectively. This highlights that during autumn and spring, the estimated WHS-attributable health burden is low compared to winter months, but relatively important in the case of LFS (See Appendix A for detailed results on sensitivity analysis scenarios).

## 4. Discussion

We calculated that each year on average, AUD$ 309 (95% CI: 194–419) million in health costs can be attributed to biomass smoke exposure in Tasmania, with the vast majority relating to WHS, although the daily impacts from LFS can be extreme during severe bushfire periods.

### 4.1. Results in Relation to Other Studies

In Tasmania WHS health impacts occur during winter months and are concentrated in the two largest cities, Hobart and Launceston, where the greatest numbers of wood heaters are located. For example, Launceston has around one third (21,800) of these appliances in Tasmania, and has historically had serious air pollution from wood smoke [22,26], although policy interventions such as educational campaigns, enforcement of environmental regulations, and wood heater changeout programs have reduced the impact [23]. We estimated that health costs attributable to WHS PM_2.5_ were over AUD$ 290 million per year, and on average represented 94.8% of all biomass smoke costs. Most of these costs were attributable to the estimated 65 premature deaths (12.5 deaths per 100,000 persons per year) which account for 1.5% of total yearly deaths in Tasmania. These results were within the range of biomass health impacts modeled in other locations globally. For example, Sarigiannis et al. [15] estimated 22 deaths per 100,000 persons per year for the 2012/2013 winter in Thessaloniki (Greece), and for 2010, Chafe et al. [52] estimated ~8.2 cases per 100,000 in Europe and ~2.9 cases per 100,000 in North America. Such variation is not surprising because wood heater impacts on air quality, and population vulnerability due to factors such as demographic structure and underlying health status, will vary from place to place.

Our estimates for LFS-associated health impacts were higher than previous estimates for other regions of Australia but similar to estimates for the US; in all cases within similar orders of magnitude. For example, we estimated that on average every year, LFS was associated with 5.3 deaths per 1,000,000 persons per year, Horsley et al. [33] estimated for Sydney an average of 3.5 premature deaths per 1,000,000 persons per year, and Borchers-Arriagada et al. [53] estimated for Western Australia an average of 1 death per 1,000,000 persons per year in an analysis restricted to days when PM10 or PM_2.5_ concentrations exceeded national air quality standards [54]. In contrast, Fann et al. [14] estimated that short-term exposure to LFS PM_2.5_ in the US was associated with 6 premature deaths per 1,000,000 persons per year, resulting in estimated costs between $US 11 and $US 20 billion per year.

While LFS impacts were lower than WHS, there is a high likelihood that these type of events will increase due to climate change [55], and public health impacts will increase substantially when large populations are exposed. Even with conservative modeling assumptions, our sensitivity analysis showed that LFS-related costs were already substantial, particularly during extreme fire years.

We found summer bushfires were much more likely to be associated with increased health impacts compared to LFS days on transition months, which are generally produced by prescribed burns. This finding contrasts with other parts of Australia, such as Sydney, where smoke from prescribed burning can be extreme and potentially associated with health impacts similar to the smoke impacts from severe bushfires in those regions. For example during May 2016, in Sydney, prescribed burning activities produced six days of clearly increased PM_2.5_ which was associated with an estimated 14 premature deaths and 87 cardiovascular and respiratory hospitalizations [56].

### 4.2. Strengths and Limitations

All modeling and health and economic impact assessment studies are subject to a range of assumptions and uncertainties about exposure assessment, health coefficients selected, and economic valuation. The main strengths of this analysis relate to the application of simple and commonly used methods for the estimation of health impacts and related costs, and the implementation of a sensitivity analysis to observe how much results could vary from the initial estimations. The health coefficients used for this study have been recommended by the World Health Organization [5] or are results of previous meta-analyses, encompassing a large body of evidence. The limitations of our analysis mainly relate to the potential misclassification of elevated PM_2.5_ days according to type of source (WHS or LFS) and the estimation of PM_2.5_ exposure, particularly during transitional months. Nevertheless, by incorporating detailed meteorological data, we were able to confidently predict pollution sources during these transition months, and results were robust across our sensitivity analyses. The exclusion of transition months from our analysis produces slight reductions of total costs for WHS but larger impacts on LFS. However, most health impacts were concentrated between May and August for WHS and during January for LFS, and therefore the potential impact of misclassification of source types during transitional months on the overall results would be minimal. To attribute PM_2.5_ exposure, we combined empirical observations with inverse distance weighting interpolation to estimate average exposure at a geographical SA2 level. While PM_2.5_ exposure could potentially be improved using other methods such as satellite imagery, ordinary kriging, or land use regression models, the dense air quality network in Tasmania provides high confidence in the exposure estimates with 89% of the population living within 5 km of a monitor.

We acknowledge some uncertainty in using yearly average health data to estimate the number of cases for each outcome, given the inherent seasonality of exposure to WHS and LFS, and the likely seasonality of health outcomes as well. Overall, it is probable that our results are an underestimation, as the bulk of health impacts have been estimated for WHS during the winter season where baseline incidence rates are likely higher than the annual averages used.

We applied a recommended VSL value to estimate mortality costs, and this method does not consider possible differences by age or health status [57]. It should be noted that VSL is not an objective representation of the monetary value of a human life, but rather represents how much individuals in a population are willing to exchange part of their wealth for changes in their mortality risk [57,58]. Despite the recognized limitations, this monetization method has been widely used to quantify and value the health impacts attributable to air pollution [14,15,53,57,59,60,61]. Furthermore, our sensitivity analysis shows that although there was some variation in total costs with different model parameterization, particularly for LFS, this does not have a large influence on daily health cost estimates, and even in conservative modeling scenarios, the estimated health costs remain substantial.

### 4.3. Policy Implications

The estimated health costs were very different for WHS and LFS, but in both cases they were quite considerable and demonstrated that substantial public health benefits and large cost savings would be possible from interventions to reduce or mitigate the impacts of the exposure. Wood heater smoke is much more amenable to direct policy intervention as demonstrated by the Launceston buy-back scheme in 2001, which was associated with reduced mortality [23]. Between July 2001 and June 2004, the AUD$ 2.05 million program helped accelerate the reduction of the proportion of homes that were primarily heated by wood [23]. With estimated yearly health costs attributable to WHS in the Launceston and north–east of AUD$ 109 million per year, it is likely that the Launceston Wood Heater Replacement Program was very cost-effective, and that considerable savings could be realized through additional interventions, given our estimates of the yearly WHS health costs to be between AUD$ 3500 and AUD$ 4600 per wood heater. Funding of replacement to low- or non-polluting heating alternatives, such as pellet burners or electric reverse cycle air conditioning, or home interventions to reduce heating demand through, for example, improved insulation, would likely result in a rapid return on investment, at least from a public health perspective.

LFS impacts during severe bushfire seasons such as those of 2016 or 2019 are harder to mitigate as there is little ability to control or minimize smoke emissions in the context of fire emergencies. However, interventions focusing on people more vulnerable to harm from air pollution through education, communication, medical management of associated health conditions, and exposure reduction can all reduce the associated harm [62]. Further, the impacts from prescribed burns during the months of March, April, and October are more amenable to intervention and mitigation through coordinated smoke management [63] and advanced communications to enable people in higher risk groups to act to reduce their exposure by sealing their homes and staying indoors, using a portable air cleaner, or moving to a location less affected by smoke during the burn off period. Additionally, with advanced warning systems, people belonging to higher risk groups may take action such as using preventive medication to reduce the health impacts from exposure to smoke [62].

Health impacts from LFS PM_2.5_, whether from a wildfire or a prescribed burn, need to be considered along with many other risk assessments that support fire risk reduction interventions. This could influence the amount of resources that are allocated towards wildfire risk reduction and how preventive interventions are implemented.

### 4.4. Unanswered Questions and Future Research

Given the substantial health burden attributable to WHS and LFS, there may be some unexplored potential to reduce smoke-related PM_2.5_ in a cost-effective manner. This could be realized by approaching each of these two sources (or types of sources) independently, or by designing an integrated strategy. An integrated strategy may shift two non-cost-effective interventions to being cost-effective when implemented simultaneously and interlinked between them.

For example, some fuel risk reduction interventions that do not produce smoke, such as mechanical thinning, landscaping, and the creation of green firebreaks have higher implementation costs than the more widely practiced fuel reduction burning [64]. However, in some situations, especially close to population centers, non-combustion strategies to management fuel could be less costly overall if the full health impacts of the intervention, including those related to smoke emissions, are taken into account [64]. Nevertheless, economic constraints on the implementation of biomass removal through mechanical thinning could be viewed as an opportunity by using the removed biomass to produce energy for residential heating [65]. Furthermore, in an area where WHS is a major concern, such as Tasmania, there may be ways of linking both of these environmental issues. One plausible solution would be to implement an integrated mechanical thinning and wood heater changeout program, in which removed biomass could be transformed to wood pellets or chips, which would ultimately be used to produce cleaner energy by using technologically sophisticated, highly efficient, and low-polluting biomass heaters such as pellet stoves [64].

Both these interventions, if implemented separately, would probably translate into relatively high initial costs for investors and individuals. Yet when integrated, the higher wildfire risk reduction costs could be offset by pellet sales, and considerable air quality improvements through reduced pollution from both WHS and LFS. This means improved population health and large health cost savings.

Taking into consideration the results of this study, we recommend that further assessments, such as a cost–benefit analysis incorporating the full health impacts, should be done to evaluate the feasibility of interventions that aim to solve environmental issues. Ideally, these assessments would include an analysis of a variety of pollution reduction strategies considering social, economic, and environmental impacts, which are evaluated and balanced using the same metrics. These types of analyses may be further used to decide on the best steps to solve the current pollution and health problem in Tasmania.

## 5. Conclusions

Our study estimates the health impacts and associated costs of population exposure to biomass smoke-related PM_2.5_, particularly that produced by landscape fires and wood heaters over a 10-year period (2010–2019) in Tasmania, the southern island state of Australian. Tasmania is characterized by having distinct seasonal pollution and temperature patterns, which are captured by relatively dense air quality and meteorological monitoring stations. Landscape fires and wood heaters are the two main sources of PM_2.5_, with only 8% attributable to other sources. We classified days as being affected by WHS or LFS during winter and summer, and then we used pollution and meteorological data to apply a random forest algorithm to predict most likely pollution source during autumn and spring. Then, we used a standard health impact assessment methodology to estimate the number of premature deaths, cardiovascular and respiratory hospital admissions, and ED visits for asthma. We estimated health costs by using VSL for mortality and national average costs for hospital admissions and ED visits. We estimated that biomass smoke was associated with 69 deaths, 86 hospital admissions, and 15 asthma ED visits each year, with over 74% of impacts attributed to WHS. This translates into average yearly costs of AUD$ 293 million for WHS and AUD$ 16 million for LFS. LFS costs increase substantially during extreme fire years, such as 2016 and 2019, reaching more than AUD$ 34 million per year. Biomass smoke pollution is a growing public health issue for landscape fire smoke and residential wood heating. With global warming, it is expected that extreme weather events, including landscape fires, will be more frequent and intense. Additionally, the use of wood for residential heating is not an issue that only affects lower and medium income countries, as it gains popularity in places such as Australia, the US and Europe. The reduction of exposure to biomass PM_2.5_, through better and innovative fire management, the replacement in the use of poorly designed highly pollutant wood heating technologies with more modern and efficient designs, and possibly the implementation of integrated strategies, has the potential to produce important and cost-effective health benefits.

## Figures and Tables

**Figure 1 ijerph-17-03264-f001:**
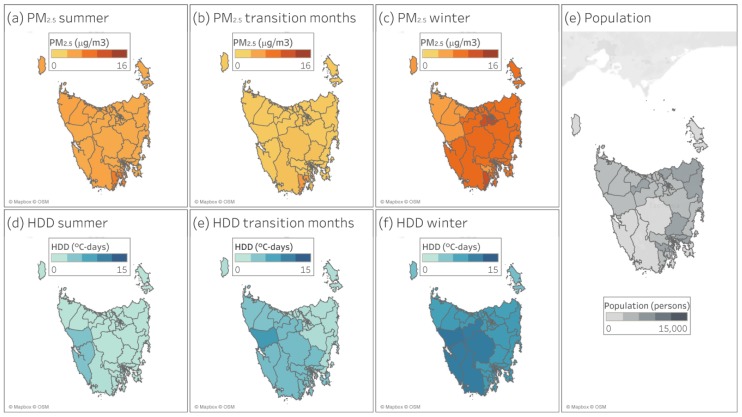
Average summer, transition, and winter months fine particulate matter (PM_2.5_) concentration and heating degree days (HDD) and exposed population.

**Figure 2 ijerph-17-03264-f002:**
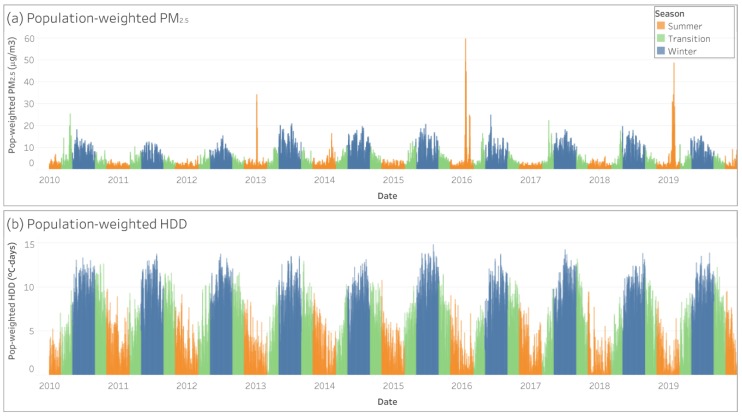
Average daily PM_2.5_ concentration (μg/m^3^) and HDD (°C-days) for the period of analysis by season.

**Figure 3 ijerph-17-03264-f003:**
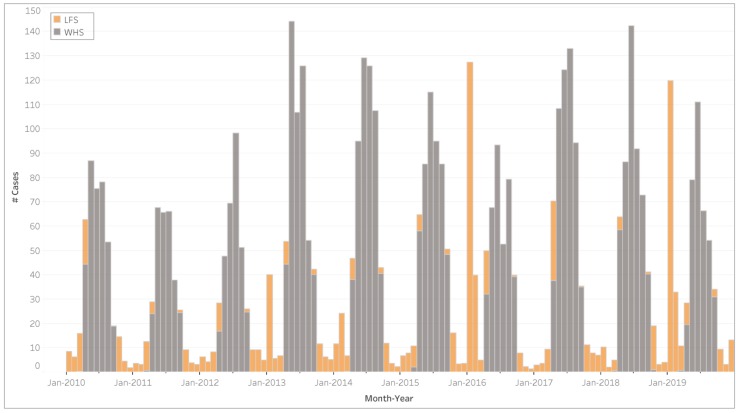
Total estimated number of cases attributable to wood heater smoke (WHS) and landscape fire smoke (LFS) by month and year.

**Figure 4 ijerph-17-03264-f004:**
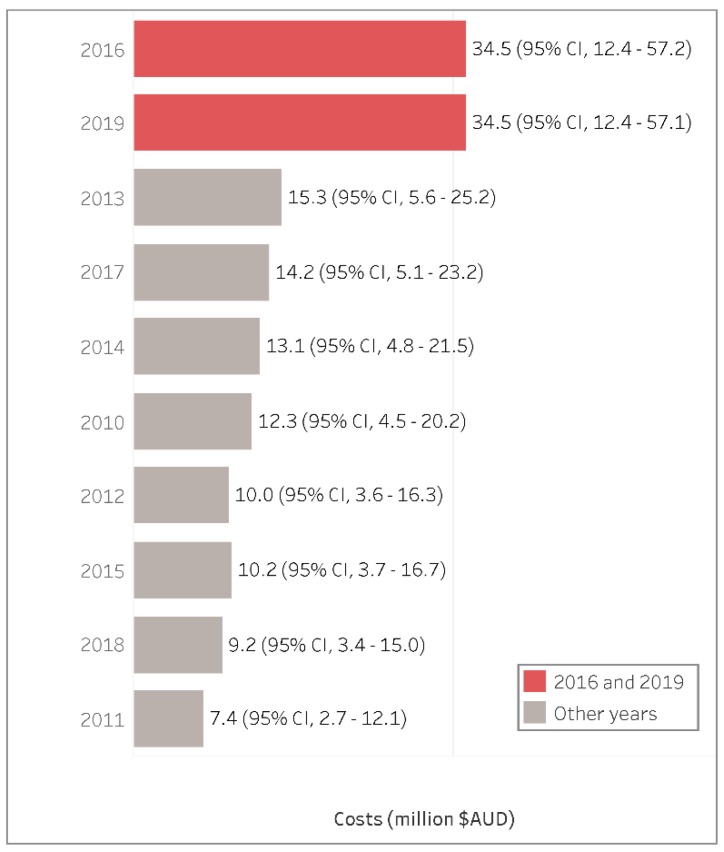
Total estimated LFS health costs per year in descending order (million AUD$ 2018).

**Table 1 ijerph-17-03264-t001:** Fine particulate matter (PM_2.5_) dose-response functions selected for wood heater smoke (WHS) and landscape fire smoke (LFS).

Effect	Cause	Age Group	Type of Smoke	Exposure	Beta	Standard Error	Increase per 10 (μg/m^3^)	Reference
Premature Mortality	All-cause	30+	WHS	Annual PM_2.5_	0.006015	0.001034	6.2%	[5]
		All	LFS	24-h PM_2.5_	0.001222	0.000393	1.2%	
Hospital Admissions	CVD	All	WHS/LFS	24-h PM_2.5_	0.000906	0.000377	0.9%	[5]
	RSP	All	WHS/LFS	24-h PM_2.5_	0.001882	0.001051	1.9%	
ED visits	Asthma	0–17	WHS	24-h PM_2.5_	0.003537	0.000862	3.6%	[42]
		18–64	WHS	24-h PM_2.5_	0.001686	0.000527	1.7%	
		0–17	LFS	24-h PM_2.5_	0.003811	0.001865	3.9%	[11]
		18–64	LFS	24-h PM_2.5_	0.007071	0.001586	7.3%	
		64+	LFS	24-h PM_2.5_	0.013994	0.002356	15%	

CVD = Cardiovascular. RSP = Respiratory. ED = Emergency Department.

**Table 2 ijerph-17-03264-t002:** Average number of days per year (total and %) and 24-h PM_2.5_ summary statistics by day type.

Day Type	Average Days per Year (number)	Average Days per Year (%)	Mean (μg/m^3^)	SD (μg/m^3^)	Max (μg/m^3^)
Unpolluted	95	26%	1.4	0.4	3
LFS	111.8	30.6%	4.2	6.3	313
WHS	158.3	43.4%	7.7	5.4	168

SD = Standard Deviation.

**Table 3 ijerph-17-03264-t003:** Estimated number of cases per health outcome and source type.

Source Type	Health Outcome	Estimated # of Cases	Estimated # of Cases per Year
LFS	All-cause Mortality	36 (95% CI: 13–58)	4 (95% CI: 1–6)
	Asthma ED Visits	76 (95% CI: 39–121)	8 (95% CI: 4–12)
	CVD Hospital Admissions	70 (95% CI: 13–128)	7 (95% CI: 1–13)
	RSP Hospital Admissions	112 (95% CI: 0–240)	11 (95% CI: 0–24)
WHS	All-cause Mortality	653 (95% CI: 420–874)	65 (95% CI: 42–87)
	Asthma ED Visits	72 (95% CI: 36–108)	7 (95% CI: 4–11)
	CVD Hospital Admissions	259 (95% CI: 49–474)	26 (95% CI: 5–47)
	RSP Hospital Admissions	416 (95% CI: 0–882)	42 (95% CI: 0–88)
Total	All-cause Mortality	688 (95% CI: 433–932)	69 (95% CI: 43–93)
	Asthma ED Visits	148 (95% CI: 74–229)	15 (95% CI: 7–23)
	CVD Hospital Admissions	329 (95% CI: 62–602)	33 (95% CI: 6–60)
	RSP Hospital Admissions	528 (95% CI: 0–1123)	53 (95% CI: 0–112)

**Table 4 ijerph-17-03264-t004:** Estimated number of cases attributable to LFS per health outcome for 2016, 2019, and average for all other years.

Health Outcome	2016	2019	Average for All Other Years
All-cause Mortality	8 (95% CI: 3–13)	8 (95% CI: 3–13)	3 (95% CI: 1–4)
Asthma ED Visits	18 (95% CI: 9–31)	18 (95% CI: 9–30)	5 (95% CI: 3–8)
CVD Hospital Admissions	15 (95% CI: 3–28)	15 (95% CI: 3–28)	5 (95% CI: 1–9)
RSP Hospital Admissions	24 (95% CI: 0–52)	24 (95% CI: 0–52)	8 (95% CI: 0–17)

**Table 5 ijerph-17-03264-t005:** Estimated total costs and costs per year by source type and health outcome.

Source Type	Health Outcome	Total Cost (AUD$ *)	Yearly Cost (AUD$/Year *)
LFS	All-cause Mortality (**)	159 (95% CI: 58–262) million	16 (95% CI: 6–26) million
	Asthma ED Visits	55,949 (95% CI: 28,370–88,766)	5595 (95% CI: 2837–8877)
	CVD Hospital Admissions	521,552 (95% CI: 97,075–957,899)	52,155 (95% CI: 9708–95,790)
	RSP Hospital Admissions	845,972 (95% CI: 0–1,817,405)	84,597 (95% CI: 0–181,741)
WHS	All-cause Mortality (**)	2929 (95% CI: 1885–3920) million	293 (95% CI: 189–392) million
	Asthma ED Visits	52,508 (95% CI: 26,071–79,259)	5251 (95% CI: 2607–7926)
	CVD Hospital Admissions	1,937,594 (95% CI: 362,145–3,542,801)	193,759 (95% CI: 36,215–354,280)
	RSP Hospital Admissions	3,149,178 (95% CI: 0–6,674,047)	314,918 (95% CI: 0–667,405)
Total	All-cause Mortality (**)	3088 (95% CI: 1943–4181) million	309 (95% CI: 194–418) million
	Asthma ED Visits	108,457 (95% CI: 54,441–168,025)	10,846 (95% CI: 5444–16,803)
	CVD Hospital Admissions	2,459,146 (95% CI: 459,220–4,500,700)	245,915 (95% CI: 45,922–450,070)
	RSP Hospital Admissions	3,995,150 (95% CI: 0–8,491,452)	399,515 (95% CI: 0–849,145)

(*) Costs presented as 2018 AUD$. (**) costs for all-cause mortality presented as million AUD$.

**Table 6 ijerph-17-03264-t006:** Estimated population (2017) and total costs by source type and region (SA4).

SA4	Population		LFS		WHS	
	Persons	%	AUD$	%	AUD$	%
Hobart	229,088	44%	63,119,001	39%	1,161,808,071	40%
Launceston and North–East	143,752	28%	48,045,137	30%	1,093,171,538	37%
West and North–West	111,259	21%	36,536,828	23%	473,598,790	16%
South–East	38,053	7%	13,016,748	8%	205,218,190	7%
**Total**	**522,152**	**100%**	**160,717,714**	**100%**	**2,933,796,590**	**100%**

SA4 = Statistical Area Level 4.

**Table 7 ijerph-17-03264-t007:** Health economic impacts associated with biomass smoke for WHS and LFS between 2010 and 2019.

Source	Indicator	All-Cause Mortality	Asthma ED Visits	CVD Hospital Admissions	RSP Hospital Admissions	TOTAL
WHS	Total Cost (AUD$)	2,928,657,309	52,508	1,937,594	3,149,178	**2,933,796,590**
	Cost per day (AUD$/WHS-day)	1,566,127	28	1036	1684	**1,568,875**
	Cost per year (AUD$/year)	292,865,731	5251	193,759	314,918	**293,379,659**
	Cost per woodstove-year (AUD$/woodstove-year)	4225	0	3	5	**4232**
LFS	Total Cost (AUD$)	159,294,240	55,949	521,552	845,972	**160,717,714**
	Cost per day (AUD$/LFS-day)	75,281	26	246	400	**75,954**
	Cost per year (AUD$/year)	15,929,424	5595	52,155	84,597	**16,071,771**

**Table 8 ijerph-17-03264-t008:** Health economic indicator ranges from sensitivity analysis.

Months Included	Source	Indicator	Main Scenario	Range
All months	WHS	Total Cost (AUD$)	2934 million	2458–3189 million
		Cost per day (AUD$/WHS-day)	1.6 million	1.4–1.7 million
		Cost per year (AUD$/year)	293.4 million	245.8–318.9 million
		Cost per woodstove-year (AUD$/woodstove-year)	4232	3545–4601
	LFS	Total Cost (AUD$)	160.7 million	138.2–270.3 million
		Cost per day (AUD$/LFS-day)	76 thousand	64–109 thousand
		Cost per year (AUD$/year)	16.1 million	13.8–27 million
Excluding months with predicted biomass smoke source	WHS	Total Cost (AUD$)	2438 million	1294–2615 million
		Cost per day (AUD$/WHS-day)	1.98 million	1.9–2.4 million
		Cost per year (AUD$/year)	243.8 million	129.4–261.5 million
		Cost per woodstove-year (AUD$/woodstove-year)	3518	1867–3773
	LFS	Total Cost (AUD$)	97.3 million	86–117.3 million
		Cost per day (AUD$/LFS-day)	81 thousand	71.5–97.6 thousand
		Cost per year (AUD$/year)	9.7 million	8.6–11.7 million

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
