# Peer review of "Health Impacts of Ambient Biomass Smoke in Tasmania, Australia"

_ijerph, 2020, doi:10.3390/ijerph17093264_

Round 1

Reviewer 1 Report

1.The abstract must clearly describe the specific contribution of the results.

2.In the abstract, especially the methodology and quantitative and qualitative results.(This article abstract presents more descriptive statements)

3.Methodology and experiments have highlights, especially PM2.5

4.Conclusions must be modified to describe or highlight the specific contribution of this article.

Author Response

Dear reviewer

Thank you so much for your suggestions. Please find our answers to each of your comments/suggestions below:

  1. The abstract must clearly describe the specific contribution of the results.

We modified the abstract, particularly added the last sentences as follows (Line 30):

“This is the first study to quantify the health impacts attributable to biomass smoke for Tasmania. We estimated substantial impacts, which could be reduced through reducing the use of highly pollutant wood heating technologies, better and innovative fire management, and possibly the implementation of integrated strategies. This would most likely produce important and cost-effective health benefits.”

  1. In the abstract, especially the methodology and quantitative and qualitative results. (This article abstract presents more descriptive statements)

The abstract was modified, and now reads as follows:

“The island state of Tasmania has marked seasonal variations of fine particulate matter (PM2.5) concentrations related to wood heating during winter, planned forest fires during autumn and spring, and bushfires during summer. Biomass smoke causes considerable health harms and associated costs. We estimated the historical health burden from PM2.5 attributable to wood heater smoke (WHS) and landscape fire smoke (LFS) in Tasmania between 2010 and 2019. We calculated the daily population level exposure to WHS and LFS-related PM2.5 and estimated to estimate the number of cases and health costs due to premature mortality, cardiorespiratory hospital admissions, and asthma ED visits. We estimated 69 deaths, 86 hospital admissions, and 15 asthma ED visits, each year, with over 74% of impacts attributed to WHS. Average yearly costs associated with WHS were of AUD$ 293 million and AUD$ 16 million for LFS. The latter increased up to more than AUD$ 34 million during extreme bushfire seasons. This is the first study to quantify the health impacts attributable to biomass smoke for Tasmania. We estimated substantial impacts, which could be reduced through replacing heating technologies, improving fire management, and possibly implementing integrated strategies. This would most likely produce important and cost-effective health benefits.”

  1. Methodology and experiments have highlights, especially PM2.5

Thank you for highlighting this. We have made substantial improvements to the manuscript, modifying the abstract, introduction (particularly objectives), methods, discussion and conclusions. We have also improved the quality of figures, and have reduced the amount of figures and tables by either deleting a few or moving to Supplementary Material. We made sure that referencing of supplementary figures and tables followed the same order as presented in the actual supplement.

  1. Conclusions must be modified to describe or highlight the specific contribution of this article.

Conclusions have been re-written and now read as follows:

“Our study estimates the health impacts and associated costs of population exposure to biomass smoke-related PM2.5, particularly that produced by landscape fires and wood heaters, for a 10 year period (2010 – 2019) in Tasmania, the southern island state of Australian. Tasmania is characterized by having distinct seasonal pollution and temperature patterns, which are captured by relatively dense air quality and meteorological monitoring stations. Landscape fires and wood heaters are the two main sources of PM2.5, with only 8% attributable to other sources. We classified days as being affected by WHS or LFS during winter and summer, and then we used pollution and meteorological data to apply a random forest algorithm to predict most likely pollution source during autumn and spring. Then, we used a standard health impact assessment methodology to estimate the number of premature deaths, cardiovascular and respiratory hospital admissions, and ED visits for asthma. We estimated health costs by using VSL for mortality and national average costs for hospital admissions and ED visits. We estimated that biomass smoke was associated with 69 deaths, 86 hospital admissions, and 15 asthma ED visits, each year, with over 74% of impacts attributed to WHS. This translates into average yearly costs of AUD$ 293 for WHS and AUD$16 for LFS. LFS costs increase substantially during extreme fire years, such as 2016 and 2019, reaching more than AUD$ 34 million per year. Biomass smoke pollution is a growing public health issue, for landscape fire smoke and residential wood heating. With global warming, it is expected that extreme weather events, including landscape fires, will be more frequent and intense. Additionally, the use of wood for residential heating is not an issue that only affects lower and medium income countries, as it gains popularity in places such as Australia, the US and Europe. The reduction of exposure to biomass PM2.5, through better and innovative fire management, the decrease in the use of highly pollutant wood heating technologies, and possibly the implementation of integrated strategies, has the potential to produce important and cost-effective health benefits.”

Reviewer 2 Report

to make it clear from the beginning that all research is based on statistics achieved by other parties/national parties in principal, that are official.

make a clear comparison between the two emission exhaust characteristics (two main sources according to the article content-comparison) - this must be science  based on clear measurements

to be more explained *we 139 estimated short term impacts using average daily exposure because there is inadequate evidence to 140 support the estimate longer term health outcomes associated with yearly average exposure* 

refresh the title of the article by combining both ideas, make it short. presently *Health impacts of ambient biomass smoke in 2 Tasmania, Australia. Comparative analysis of 3 landscape fires and wood heating* is quite long.

make a clear difference in the *Unanswered questions and future research* 

Author Response

Dear reviewer,

Thank you so much for your review. Please find below our answers to each one of your comments/suggestions:

  1. to make it clear from the beginning that all research is based on statistics achieved by other parties/national parties in principal, that are official.

Last paragraph of introduction, including objectives, was re-written to include a statement on the data we used for this study:

“Previous studies have used Tasmania as a setting, and have characterized air pollution [18,19,22–24], assessed public health interventions to reduce WHS emissions [21], and quantified the association between biomass smoke and health outcomes [5,20]. No study has quantified the health burden attributable to biomass smoke-related PM2.5 in Tasmania, and compared the relative contribution of LFS and WHS. We used publicly available demographic, health and air quality data, to estimate and compare health impacts and health costs attributable to the two most important sources of air pollution in Tasmania, wood heaters and landscape fires, between 2010 and 2019.”

  1. make a clear comparison between the two emission exhaust characteristics (two main sources according to the article content-comparison) - this must be science based on clear measurements

We have modified the introduction, and first sentences now include a description of how WHS and LFS are different in composition, but we highlight that we are focusing in PM2.5:

“Smoke from biomass combustion, including wood heater smoke (WHS) and landscape fire smoke (LFS), is composed of a complex blend of pollutants such as particulate matter, carbon monoxide and volatile organic gases [1,2]. WHS is produced by emissions from a myriad of residential heating technologies such as wood or pellet stoves, biomass boilers, and open fire places. There is a great variation in the physicochemical properties of particles that are emitted, and they depend on the type of technology, fuel conditions, and fuel types, among others [3]. Likewise, the composition of LFS varies according to vegetation type, climate conditions, and intensity of burn [4]. Short- and long-term exposure to particulate matter, specifically the fine fraction that contains particles of size up to 2.5 mm (PM2.5), has been clearly linked to several health problems, including premature mortality, hospital admissions, and emergency department (ED) visits [5]. Multiple studies have investigated these exposure-outcome relationships and estimated the health impacts, particularly for WHS and LFS [6–12].”

  1. to be more explained *we 139 estimated short term impacts using average daily exposure because there is inadequate evidence to 140 support the estimate longer term health outcomes associated with yearly average exposure*

That section was re-written to better explain difference between long- and short- term exposure natures of WHS and LFS. It reads as follows:

“Timeframes of exposure to WHS-related PM2.5 and LFS-related PM2.5 are different by nature, with population exposure to WHS happening every year throughout winter months, and exposure to LFS happening sporadically and during a shorter duration (i.e. days) mainly during summer months. Given this, we selected different dose-response functions to assess impact of smoke exposure on premature mortality. In the case of WHS, we assessed long term impacts using average annual exposure, for which the relationships have been characterized [3]. For LFS, we estimated impacts on premature mortality by using average daily exposure, as there is no available evidence on the association between premature mortality and long-term exposure to LFS [39], and this a sporadic rather than a chronic phenomena in Tasmania. For hospital admission and ED visits, we used average daily exposure [3,9,40]. We selected the health coefficients presented in Table 1, and considered uncertainty associated with selected coefficients, to obtain the health impacts’ 95% confidence intervals.”

  1. refresh the title of the article by combining both ideas, make it short. presently *Health impacts of ambient biomass smoke in 2 Tasmania, Australia. Comparative analysis of 3 landscape fires and wood heating* is quite long.

Changed title to the following: “Health impacts of ambient biomass smoke in Tasmania, Australia”

  1. make a clear difference in the *Unanswered questions and future research*

We have modified that section. Firstly, we introduce that the estimated health burden is substantial and there is an opportunity. Then, we give an example of an integrated intervention, which tackles LFS and WHS in an interlinked manner. Finally, we conclude that further studies, particularly cost-benefit analysis of different pollution reduction strategies, should be done, to better inform policy.

Reviewer 3 Report

Revision of the manuscript: Health impacts of ambient biomass smoke in Tasmania, Australia. Comparative analysis of landscape fires and wood heating

The present manuscript reports a well-designed and structured work related with the assessment of health and economic impacts of ambient biomass smoke, from both wood heating smoke and landscape fires smoke, in the Tasmanian population. A large amount of environmental data was collected between 2010-2019 concerning the concentrations of ambient PM2.5.  A complete and extensive health and economic risk assessments were performed by the authors. However, the way the produced data is presented should be carefully revised. In its present form the manuscript as 10 tables and 8 figures in approximately 17 pages (excluding the abstract and the references pages).  Moreover, the supplementary material also has 8 figures and 3 tables.

Thus, I recommend the reorganization of information in order to highlight the profile of distribution of ambient PM2.5 over the selected period (2010-2019) as well as the wood heating smoke and landscape fires smoke exposure profiles and their respective health and economic costs (e.g., as presented in Figure 2). A consistent approach will allow a better comparison of the presented data over the selected period. Tables 1 and 3 can be moved to the supplemented material.

The quality of some images (e.g., Figure 1) can be improved by increasing the letter size and quality. Authors should adjust the year-axis of Figure 2 because it seems that data from the year 2020 is presented.

Data reported in Table 4 would be much more meaningfully and expressive for comparison if presented by year (2010-2019). Also, data from Figure 3 should be presented per year to allow a better comparison.

Please see some specific suggestions:

The abstract should indicate the period of time covered by the study.

Introduction

Lines 46-49: Authors should revise the sentence in order to clarify and if possible explain based on the reported fires events rather than only the presented range of mean LFS- derived PM2.5 values (0.6-1.1 µg/m3) as causing the reported "health costs".

Objectives should be rewritten to became more specific and explaining the different approaches selected by the authors to evaluate the health and economic impacts in the exposed populations.

Materials and Methods

Authors should give information regarding i) the number of monitoring stations used in to collect PM2.5 levels and ii) the background influence of the selected monitoring stations, i.e., rural, sub-urban, urban with or without background or traffic influence.

Lines 99-101: The information present in the supplementary material ("Air pollution generated by wood heaters produces characteristic seasonal and diurnal patterns with a large peak overnight, and smaller peak in the early morning. This pattern common in Tasmania throughout the cooler months [3–6] (Figure S1).")  is relevant and should be in the manuscript; authors should refer to Figure 1S.

Supplementary material is not presented in the text by the order of first citation, e.g., Figure S3 of the supplementary material is the first cited figure in the manuscript. Also, no significant information can be retrieved from Figure S6, since no difference was observed in the monthly levels of estimated background PM2.5 fraction.

Author Response

Dear reviewer,

Thank you very much for your review. Please find below responses to each of your suggestions/comments:

Revision of the manuscript: Health impacts of ambient biomass smoke in Tasmania, Australia. Comparative analysis of landscape fires and wood heating

The present manuscript reports a well-designed and structured work related with the assessment of health and economic impacts of ambient biomass smoke, from both wood heating smoke and landscape fires smoke, in the Tasmanian population. A large amount of environmental data was collected between 2010-2019 concerning the concentrations of ambient PM2.5. A complete and extensive health and economic risk assessments were performed by the authors. However, the way the produced data is presented should be carefully revised. In its present form the manuscript as 10 tables and 8 figures in approximately 17 pages (excluding the abstract and the references pages). Moreover, the supplementary material also has 8 figures and 3 tables.

We revised all tables and figures. After incorporating reviewer’s comments, the manuscript now has 8 tables and 4 figures.

Thus, I recommend the reorganization of information in order to highlight the profile of distribution of ambient PM2.5 over the selected period (2010-2019) as well as the wood heating smoke and landscape fires smoke exposure profiles and their respective health and economic costs (e.g., as presented in Figure 2). A consistent approach will allow a better comparison of the presented data over the selected period. Tables 1 and 3 can be moved to the supplemented material.

Tables 1 and 3, moved to supplementary material as Tables S1 and Tables S2, and referenced in text.

The quality of some images (e.g., Figure 1) can be improved by increasing the letter size and quality.

We increased letter size and legend size. We also added a border to each panel, to improve the look. Other figures were revised to see if they required improved quality.

Authors should adjust the year-axis of Figure 2 because it seems that data from the year 2020 is presented.

We adjusted axis to start on 01/01/2010 and end 12/31/2019. We also modified font size to improve readability

Data reported in Table 4 would be much more meaningfully and expressive for comparison if presented by year (2010-2019).

We modified this table as follows. Changed # days to ‘Average days per year’, calculating the average across all years (10) and SA2s. We also added the column ‘Average % days per year’ to show how average years distribute by day type across all years and SA2s. Additionally, we included a similar table with detail per year in Supplementary Material (Table S5)

Also, data from Figure 3 should be presented per year to allow a better comparison.

Figure 3 modified to present #cases per month-year. This new figure, clearly shows increase in #cases during January 2016 and January 2019, so Figure 4 which had that objective, was deleted.

Please see some specific suggestions:

The abstract should indicate the period of time covered by the study.

We changed the following sentence in abstract to include study period: “We aimed to estimate the historical health impacts and health costs from PM2.5 attributable to wood heater smoke (WHS) and landscape fire smoke (LFS) in Tasmania between 2010 and 2019”

Introduction

Lines 46-49: Authors should revise the sentence in order to clarify and if possible explain based on the reported fires events rather than only the presented range of mean LFS- derived PM2.5 values (0.6-1.1 µg/m3) as causing the reported "health costs".

Fann et al. (2017) do not present results with detail on fire events, but rather show aggregated and yearly results for the period 2008-2012. We modified the sentence into two, to better explain their results:

“In the US, Fann et al [7] estimated that for the period 2008-2012 health costs of short term exposure to wildfire smoke PM2.5 ranged between US$ 11 and US$ 20 billion per year, while costs associated with long term exposure ranged between US$ 76 and US$ 130 billion per year (2010 $US). The authors observed that a relatively small number of states were highly affected by fires during this period, and that the population-weighted annual mean LFS - attributable PM2.5 ranged between 0.6 to 1.1 mg/m3 [7].”

Objectives should be rewritten to become more specific and explaining the different approaches selected by the authors to evaluate the health and economic impacts in the exposed populations.

Last paragraph of introduction, including objectives, was re-written and now reads as follows:

“Previous studies have used Tasmania as a setting, and have characterized air pollution [18,19,22–24], assessed public health interventions to reduce WHS emissions [21], and quantified the association between biomass smoke and health outcomes [5,20]. No study has quantified the health burden attributable to biomass smoke-related PM2.5 in Tasmania, and compared the relative contribution of LFS and WHS. We used publicly available demographic, health and air quality data, to estimate and compare health impacts and health costs attributable to the two most important sources of air pollution in Tasmania, wood heaters and landscape fires, between 2010 and 2019.”

Materials and Methods

Authors should give information regarding i) the number of monitoring stations used in to collect PM2.5 levels and ii) the background influence of the selected monitoring stations, i.e., rural, sub-urban, urban with or without background or traffic influence.

We included the following sentences in section ‘Materials and Methods/ PM2.5 exposure and identification of WHS and LFS days’:

“Tasmania air quality monitoring network is relatively dense, with 17 monitors active in 2010 and 35 in 2019. Likewise, the BOM meteorological network in Tasmania is composed of 58 active monitors in 2010 and 57 in 2019.”

With regards to background influence, see last paragraph of section ‘Materials and Methods/ PM2.5 exposure and identification of WHS and LFS days’. It explains contribution and seasonality of biomass smoke, and we added the following sentence:

“Ambient PM2.5 in Tasmania is highly dominated by biomass (wood heater and landscape fires) smoke, and in some locations less than 8% would be attributable to other sources such as vehicle emissions, local industry and other sources of fine aerosols [13]

Lines 99-101: The information present in the supplementary material ("Air pollution generated by wood heaters produces characteristic seasonal and diurnal patterns with a large peak overnight, and smaller peak in the early morning. This pattern common in Tasmania throughout the cooler months [3–6] (Figure S1).") is relevant and should be in the manuscript; authors should refer to Figure 1S.

That sentence has been added in main text, with slight modifications, and Figure S1 is referenced: “Air pollution generated by wood heaters follows a common pattern throughout the cooler months, with characteristic seasonal and diurnal patterns with a large peak overnight, and smaller peak in the early morning [3–6] (see Figure S1 in Supplementary Material)”

Supplementary material is not presented in the text by the order of first citation, e.g., Figure S3 of the supplementary material is the first cited figure in the manuscript. Also, no significant information can be retrieved from Figure S6, since no difference was observed in the monthly levels of estimated background PM2.5 fraction.

Manuscript has been revised, and figures in supplement follow same order as citation in main text. Figure S6 has been deleted from supplementary material.

Reviewer 4 Report

By comparing the LFS and WHS related health impact, this paper provide a novel angle for the biomass smoke related air quality analysis. The manuscript is well developed and the presentation is clear. I think this paper suits well for the scope of IJERPH. I would recommend it for publication after minor revision.

My only concern are regarding to the method sector. Although some citation is mention for the SA2, IDW and random forest algorithm, it would be better to have some basic details provide/summarized in the manuscript. So that the reader could have more straight forward understanding about the method used for LFS/WHS identification.

Author Response

Dear reviewer,

Thank you so much for reviewing our manuscript. Please find below our response to your comments/suggestions:

By comparing the LFS and WHS related health impact, this paper provides a novel angle for the biomass smoke related air quality analysis. The manuscript is well developed and the presentation is clear. I think this paper suits well for the scope of IJERPH. I would recommend it for publication after minor revision.

My only concern are regarding to the method sector. Although some citation is mention for the SA2, IDW and random forest algorithm, it would be better to have some basic details provided/summarized in the manuscript. So that the reader could have more straight forward understanding about the method used for LFS/WHS identification.

We modified sentences referring to SA2 and IDW methods:

Daily PM2.5 exposure, daily HDD, minimum daily temperature, and maximum daily temperature were interpolated at a Statistical Area Level 2 (SA2), a geographical area defined by the Australian Bureau of Statistics [27], that is characterized by having an average population of 10,000, ranging between 3,000 and 25,000 persons. We used an inverse distance weighting (IDW) method [28], a spatial interpolation algorithm which uses observations at known locations (e.g. air quality at monitoring stations), to calculate unknown values at other places by giving more importance (weight) to known values that are closer compared to those that are farther away. We estimated daily averages (PM2.5, HDD, temperature) at each SA2 by considering only AQ monitoring stations or BOM meteorological stations that were within a 100 km radius from the SA2 centroid.

With regards to the random forest method, we already present the information below. If required, we could present some statistics associated with the random forest models, to show goodness of fit, prediction error, etc. in the supplementary material.

“Air pollution from wood heaters and landscape fires has characteristic seasonal and daily temporal patterns which make discerning the source of air pollution in Tasmania straightforward for the summer and winter periods (see Figures S2 and S3 in supplementary material). Ambient PM2.5 in Tasmania is highly dominated by biomass (wood heater and landscape fires) smoke, and in some locations less than 8% would be attributable to other sources such as vehicle emissions, local industry and other sources of fine aerosols [20]. Air pollution generated by wood heaters follows a common pattern throughout the cooler months, with characteristic seasonal and diurnal patterns with a large peak overnight, and smaller peak in the early morning [18–20,23]. However, the transition months during autumn and spring potentially have both sources, depending on daily weather conditions which might either favor wood heater use, or landscape fires. For transition months (March, April, September, October), we predicted the most probable source by using a random forest machine learning algorithm known as random forest. This type of algorithm applies random sampling over a set of observations with known categories or classifications to train a model, and later uses this model to predict over observations with unknown categories [34]. We trained a model using known source categories during summer (LFS) and winter (WHS), and applied it to days during transition months, using the following explanatory variables: geographic location by statistical area (SA2), year, month, day, daily PM2.5 average, daily HDD average, day of the week, minimum daily temperature, and maximum daily temperature.

We evaluated the sensitivity of our results to the following assumptions:

  • The PM2.5 threshold used to identify LFS summer days (90th vs 99th percentile)
  • The months considered as start and end of winter, and
  • The consideration of sources allocated during the transition months (March, April, September, October) through the random forest method

Reviewer 5 Report

Please see attached the comments and suggestions

Author Response

Dear reviewer

Thank you very much for reviewing our manuscript. Please find below our responses to each one of your comments/suggestions:

Comments to the authors of the paper “Health impacts of ambient biomass smoke in 2 Tasmania, Australia. Comparative analysis of 3 landscape fires and wood heating”

General comments:

In all sections, more attention should be paid to clearly presenting the hypotheses, the data treatment and related uncertainties, the representativeness of the results obtained in relation to the conclusions drawn, and the limitations.

Thank you for highlighting this. We have made substantial improvements to the manuscript, modifying the abstract, introduction (particularly objectives), methods, discussion and conclusions. We have also improved the quality of figures, and have reduced the amount of figures and tables by either deleting a few or moving to Supplementary Material. We made sure that referencing of supplementary figures and tables followed the same order as presented in the actual supplement.

I suggest merging the Results and Discussion sections.

Thank you for this suggestion, but we are following research manuscript sections as required by IJERPH instructions to authors (https://www.mdpi.com/journal/ijerph/instructions#manuscript). These sections are: Introduction, Materials and Methods, Results, Discussion, Conclusions, Patents.

The Conclusion section should be substantially improved.

Conclusions have been re-written and now read as follows:

“Our study estimates the health impacts and associated costs of population exposure to biomass smoke-related PM2.5, particularly that produced by landscape fires and wood heaters, for a 10 year period (2010 – 2019) in Tasmania, the southern island state of Australian. Tasmania is characterized by having distinct seasonal pollution and temperature patterns, which are captured by relatively dense air quality and meteorological monitoring stations. Landscape fires and wood heaters are the two main sources of PM2.5, with only 8% attributable to other sources. We classified days as being affected by WHS or LFS during winter and summer, and then we used pollution and meteorological data to apply a random forest algorithm to predict most likely pollution source during autumn and spring. Then, we used a standard health impact assessment methodology to estimate the number of premature deaths, cardiovascular and respiratory hospital admissions, and ED visits for asthma. We estimated health costs by using VSL for mortality and national average costs for hospital admissions and ED visits. We estimated that biomass smoke was associated with 69 deaths, 86 hospital admissions, and 15 asthma ED visits, each year, with over 74% of impacts attributed to WHS. This translates into average yearly costs of AUD$ 293 for WHS and AUD$16 for LFS. LFS costs increase substantially during extreme fire years, such as 2016 and 2019, reaching more than AUD$ 34 million per year. Biomass smoke pollution is a growing public health issue, for landscape fire smoke and residential wood heating. With global warming, it is expected that extreme weather events, including landscape fires, will be more frequent and intense. Additionally, the use of wood for residential heating is not an issue that only affects lower and medium income countries, as it gains popularity in places such as Australia, the US and Europe. The reduction of exposure to biomass PM2.5, through better and innovative fire management, the decrease in the use of highly pollutant wood heating technologies, and possibly the implementation of integrated strategies, has the potential to produce important and cost-effective health benefits.”

Specific comments:

Line 24-25: the cost due to LFS is a bit confusing. Please rewrite these 2 sentences

These two paragraphs now read as follows: “We estimated yearly average costs of AUD$ 293 million for WHS and AUD$ 16 million for LFS. In the case of LFS, these costs increase substantially during extreme bushfire seasons, reaching more than AUD$ 34 million per year”

Line 25-28: the phrases seem more introductory, I suggest changing the order of the last sentences.

“Smoke pollution causes significant health harms and associated cost” replaced a similar sentence early in the abstract.

Last sentence changed to the following: “The reduction of exposure to biomass PM2.5 in Tasmania has the potential to produce important and cost-effective health benefits”. This was a conclusion of the present study, where estimated costs were relevant.

Line 36: it is too strong to link the health problems only to biomass combustion.

First three sentences were re-written to go from global to specific: 1) biomass combustion produces pollution, 2) PM2.5 (one of the pollutants from 1)) has been clearly linked to health outcomes, 3) Some studies have linked these pollutants particularly to WHS and LFS. It now reads as follows:

Biomass combustion, including wood heater smoke (WHS) and landscape fire smoke (LFS), is responsible for the release of a complex blend of pollutants such as particulate matter, carbon monoxide and volatile organic gases [1][2]. Short- and long-term exposure to particulate matter, specifically the fine fraction that contains particles of size up to 2.5 mm (PM2.5), has been clearly linked to several health problems, including premature mortality, hospital admissions, and emergency department (ED) visits [3]. Multiple studies have investigated these exposure-outcome relationships and estimated the health impacts, particularly for WHS and LFS [4–10].

Line 86: please explain BOM

BOM is an acronym to Bureau of Meteorology; this was added in line 81 after the full name.

Line 128-143: both WHS and LFS depend of the season, but you have only yearly average data in terms of health. Please add some more information regarding the uncertainties due to these aspects.

Table 1 has been moved to the supplementary material, but we have highlighted this in the Discussion. We have included the following paragraph in the ‘Discussion/Strengths and limitations’ section:

“We acknowledge some uncertainty in using yearly average health data to estimate the number of cases for each outcome, given the inherent seasonality of exposure to WHS and LFS, and the likely seasonality of health outcomes as well. Overall, it is probable that our results are an underestimation, as the bulk of health impacts have been estimated for WHS during the winter season where baseline incidence rates are likely higher than the annual averages used.”

Line 178: please add more information on the figure, e.g. HDD colour classes

Colour classes are presented for both PM2.5 and HDD, just above the Tasmania maps. Nevertheless, this figure was improved by increasing font size in panel titles and colour legends, and colouring panel borders.

Line 192: please add the units for PM2.5 concentration. Also I suggest adding the average percentage of days per year with unpolluted, WHS and LFS in order to be easier to read.

We modified this table a bit. Changed # days to ‘Average days per year’, calculating the average across all years (10) and SA2s. We also added the column ‘Average % days per year’ to show how average years distribute by day type across all years and SA2s. Additionally, we included a similar table with detail per year in Supplementary Material (Table S5)

Line 207-213: this paragraph need to be rewritten; now for example the word estimated is used twice in one line.

Paragraph has been re-written. Also, Figure 4 was deleted, as year was included in Figure 3. Now, paragraph reads as follows:

“For 2016 and 2019 we estimated 8 (95% CI, 3 - 13) premature deaths, 18 (95% CI, 9 - 31) asthma ED visits, 15 (95% CI, 3 - 28) CVD hospital admissions, and 24 (95% CI, 0 - 52) RSP hospital admissions (Table 6). The exclusion of those two years makes average yearly number of cases drop to 3 (95% CI, 1 – 4), 5 (95% CI, 3 – 8), 5 (95% CI, 1 – 9), 8 (95% CI, 0 – 17) for all-cause mortality, asthma ED visits, CVD hospital admissions and RSP hospital admissions respectively.”

Line 220: please explain more the figure.5

Figure 5 has been deleted, as the information presented here (identification of months with highest cases attributable to LFS) was duplicated from Figure 3. We moved the sentence referencing Figure 5, to the paragraph just before Figure 3, and now reads as follows:

On average, the number of cases attributable to LFS were mostly concentrated in January, followed by February, April and October.”

Line 256: I suggest moving figure 7 to the supplementary and leave here only the outcomes.

Figure 7 has been moved to supplement, but still referenced within text.

Line 266: Please add more information regarding the alternative scenarios.

Results section for sensitivity analysis was re-written, and now reads as follows:

Table 7 provides results for the different health economic indicators (defined in Supplementary Table S2). We present two broad groups, one including all months, and the other excluding months which had their pollution source predicted through a random forest algorithm. We present the range of variation for the selected indicators as a result of varying the PM2.5 threshold used to indentify LFS summer days, and the months used to define summer and winter.

Costs attributable to LFS vary considerably between $AUD 13.8 million and $AUD 27 million per year, equivalent to between $AUD 64,000 and $AUD 109,000 per LFS day. The lower variation in the average per day costs is due to the inclusion of a lower number of LFS days in the lower cost scenario. The lowest costs were estimated when the 99th percentile of historical PM2.5 daily averages was used as a threshold to identify LFS summer days and winter was defined between May and July. On the other hand, the highest costs were estimated when the 75th percentile was used to define LFS summer days and winter only included June and July. In the case of WHS, results were less sensitive, ranging between $AUD 245.8 million and $AUD 318.9 million, equivalent to between $AUD 1.4 million to $AUD 1.7 million per WHS-day, or between $AUD 3,545 and $4,600 per woodstove-year. The highest cost was obtained when threshold for identifying an LFS summer day was the 75th percentile, and winter included months between May and July. The lowest WHS costs were estimated when we used the 99th percentile threshold for LFS identification, but winter was only defined by June and July. When excluding months with predicted biomass smoke source total and yearly costs were reduced by 17% and 39% for WHS and LFS respectively. This highlights that during autumn and spring, the estimated WHS-attributable health burden is low compared to winter months, but relatively important in the case of LFS. (See Table S6 and Table S7 for detailed results on sensitivity analysis scenarios).”

Additionally, Figure 8 was deleted as it was presenting duplicate results also shown in Table 10 (now Table 7).

Line 396-406: please rewrite this paragraph emphasize the main finding in the manuscript, now it contains only general thoughts; moreover it is not appropriate to have multiple references in this section.

Conclusions have been re-written and now read as follows:

“Our study estimates the health impacts and associated costs of population exposure to biomass smoke-related PM2.5, particularly that produced by landscape fires and wood heaters, for a 10 year period (2010 – 2019) in Tasmania, the southern island state of Australian. Tasmania is characterized by having distinct seasonal pollution and temperature patterns, which are captured by relatively dense air quality and meteorological monitoring stations. Landscape fires and wood heaters are the two main sources of PM2.5, with only 8% attributable to other sources. We classified days as being affected by WHS or LFS during winter and summer, and then we used pollution and meteorological data to apply a random forest algorithm to predict most likely pollution source during autumn and spring. Then, we used a standard health impact assessment methodology to estimate the number of premature deaths, cardiovascular and respiratory hospital admissions, and ED visits for asthma. We estimated health costs by using VSL for mortality and national average costs for hospital admissions and ED visits. We estimated that biomass smoke was associated with 69 deaths, 86 hospital admissions, and 15 asthma ED visits, each year, with over 74% of impacts attributed to WHS. This translates into average yearly costs of AUD$ 293 for WHS and AUD$16 for LFS. LFS costs increase substantially during extreme fire years, such as 2016 and 2019, reaching more than AUD$ 34 million per year. Biomass smoke pollution is a growing public health issue, for landscape fire smoke and residential wood heating. With global warming, it is expected that extreme weather events, including landscape fires, will be more frequent and intense. Additionally, the use of wood for residential heating is not an issue that only affects lower and medium income countries, as it gains popularity in places such as Australia, the US and Europe. The reduction of exposure to biomass PM2.5, through better and innovative fire management, the decrease in the use of highly pollutant wood heating technologies, and possibly the implementation of integrated strategies, has the potential to produce important and cost-effective health benefits.”

Round 2

Reviewer 3 Report

The manuscript was improved and corrections were made.

Figures S2 and S6 are not mentioned in the manuscript. After these small correction, I recomment this manusccript to be accepted for publication. 

Reviewer 5 Report

An important improvement has been done from the last version. I consider that the manuscript can be accepted for publication.